# An Automatic Identification Method of Crested Ibis (*Nipponia nippon*) Habitat Based on Spatiotemporal Density Detection

**DOI:** 10.3390/ani12172220

**Published:** 2022-08-29

**Authors:** Xian Jiang, Tingdong Yang, Dongping Liu, Yili Zheng, Yan Chen, Fan Li

**Affiliations:** 1Institute of Forest Resource Information Techniques, Chinese Academy of Forestry, Beijing 100091, China; 2Key Laboratory of Forestry Remote Sensing and Information System, National Forestry and Grassland Administration, Beijing 100091, China; 3Key Laboratory of Forest Protection of National Forestry and Grassland Administration, Ecology and Nature Conservation Institute, Chinese Academy of Forestry, Beijing 100091, China; 4Environment and Nature Conservation, Chinese Academy of Forestry, Beijing 100091, China; 5School of Technology, Beijing Forestry University, Beijing 100013, China

**Keywords:** crested ibis, habitat, overnight site, foraging site, spatial density, temporal density

## Abstract

**Simple Summary:**

The trajectory data of crested ibis (*Nipponia nippon Temminck, 1835*) have been obtained by HQBG3621L backpack-style tracker. By combining the spatial and temporal features of the trajectory data, an improved spatiotemporal clustering-based DBSCAN method was adopted to detect crested ibis’s stopping points and identify the crested ibis’s habitat. The clustering results are consistent with those from remote sensing images and field surveys.

**Abstract:**

To address the current challenges of the heavy workload, time-consuming nature and labor-intensiveness involved in existing crested ibis’s (*Nipponia nippon*
*Temminck, 1835*) habitat identification approaches, this paper proposes an automatic habitat identification method based on spatiotemporal density detection. With consideration of the characteristics of the crested ibis’s trajectory data, such as aggregation, repeatability, and uncertainty, this method achieves detecting the crested ibis’s stopping points by using the spatial characteristics of the trajectory data. On this basis, an improved spatiotemporal clustering-based DBSCAN method is proposed in this paper, incorporating temporal characteristics of the trajectory data. By combining the spatial and temporal features, the proposed method is able to accurately identify the roosting and foraging sites among the crested ibis’s stopping points. Supported by remote sensing images and field investigations, it was found that the method proposed in this paper has a good clustering effect and can effectively identify the crested ibis’s foraging sites and overnight roosting areas. Specifically, the woodland, farmland, and river areas are the common foraging sites for the crested ibis, while the woodland with large trees is their common overnight site. Therefore, the method proposed in this paper can provide technical support for identifying and protecting the crested ibis’s habitats.

## 1. Introduction

The crested ibis (*Nipponia nippon Temminck, 1835*) is one of the national Class I key protected animals in China and is an endangered bird worldwide [1,2]. Since the 20th century, numerous institutes have carried out continuous protection activities and research on the crested ibis [3,4,5,6,7]. Through joint efforts, this endangered wild bird species [8,9] has been successfully protected, with a thousand-fold increase in its population. However, with the population growth and limited habitat capacity, the reproductive efficiency of the crested ibis in the core nesting area has been greatly affected [10,11,12,13]. Therefore, exploring the characteristics and selection rules of habitats such as foraging sites and overnight sites can provide a scientific basis for evaluating the living conditions of crested ibis’ so as to formulate appropriate protection and management measures.

The method used to identify the crested ibis’s habitats mainly involves field research and navigation-based positioning. The field visit and survey include baseline investigation and follow-up surveys. The living area of crested ibis’ is monitored through daily patrol of their key activity areas, breeding monitoring, and on-site investigation of their overnight roosting areas [14]. However, with the rapid growth of its population and sharp expansion of commonly used habitats, monitoring the crested ibis’s stopping points (such as overnight sites and foraging sites) by field surveying faces significant challenges, including heavy workload, time-consuming and labor-intensive. In addition, it is impossible to obtain the trajectory of their real-time movements and potential stopping points, resulting in failure to protect their habitats in a timely and effective manner. The development of wearable GPS transmitters in navigation and positioning systems provides an efficient approach for the real-time acquisition of trajectory data.

The trajectory data of the crested ibis represents the paths they follow spatially and over time. Each data point in the trajectory represents its spatial position at a certain time. Existing trajectory data mining methods mainly focus on the geometric characteristics of the trajectory without consideration of the background geographic information. Based on the trajectory data of the crested ibis, analyzing the resource status of the habitat has great significance for the protection of their habitats [13,15].

Generally, researchers use cluster analysis to study the rules of animal behavior and have achieved relatively good results [16]. However, due to disturbance from humans, natural enemies, extreme weather, and other factors, the monitoring data of crested ibis activities shows uneven density, and the traditional clustering-based method is hard to accurately identify specific behaviors, such as overnight roosting and foraging due to the unclear definition of each cluster [17,18,19].

Therefore, in view of the characteristics of the crested ibis’s trajectory data, such as aggregation, repeatability, and uncertainty, this paper proposes a habitat identification method based on spatiotemporal density. First of all, according to the spatial characteristics of the trajectory data, stopping points of the crested ibis are identified; then, based on the temporal characteristics of the trajectory data, an improved spatiotemporal clustering-based DBSCAN method is proposed to achieve accurate identification of the overnight sites and foraging sites among all stopping points, so as to intuitively analyze and visualize the temporal and spatial distribution of the crested ibis’s habitats, providing reference and decision-making support for habitat protection.

## 2. Materials and Methods

### 2.1. Overview of the Study Area

The study area is located in the Crested Ibis National Nature Reserve (107°21′–107°44′ E, 33°44′–33°35′ N) in Yangxian County, Hanzhong City of Shaanxi Province. The experimental area is situated in a transitional zone from mountains to low mountains and hills, with a total area of 37,549 hectares. The change in altitude presents an obvious vertical gradient. The lowest altitude is 500 m and the highest altitude reaches 2900 m. The area belongs to the transitional climatic zone from the warm and humid to the northern subtropical zone. The average temperature is 12–14 °C, and the annual precipitation is 900–1000 mm. There are various types of land cover in the study area, including rice paddies, reservoirs, rivers, and farmlands in valleys.

### 2.2. Collection and Preprocessing of Trajectory Data

The crested ibis’s trajectory was tracked by using the HQBG3621L backpack-style tracker (provided by Hunan Global Letter Technology Co., Ltd., Changsha, China), which is about 79 mm long, 23 mm wide, 30 mm high, and weighs about 26–32 g. The backpack-style tracker was put on three young crested ibises during the 2015 breeding season. These young crested ibises came from different nests in the Crested Ibis National Nature Reserve in Yangxian County, Shaanxi Province. The tracker collects the crested ibis’s trajectory data in real-time through the combination of Beidou and GPS. The tracker has a battery life of up to five years when the average sunshine duration is greater than 1 h. The sampling frequency of the tracker is 1 h, and the collected data is transmitted to the server through GPRS and stored in the local storage card at the same time, to ensure safe storage of trajectory data. As shown in Table 1, the trajectory data mainly includes equipment terminal number, time, longitude, latitude, speed, altitude, accuracy, and heading direction (true north is 0).

During long-term operation, due to abnormal conditions such as insufficient power supply of the tracker, extreme weather, and occlusion by obstacles, the collected data may have problems such as data loss and low accuracy. For data with a single loss rate of less than 10%, the data is filled by cubic spline interpolation; for data with a single loss rate greater than 10%, or data with an accuracy level of C or D, it should be directly eliminated.

In addition, if the speed is not “0”, indicating that the crested ibis is flying. Such data should not be considered a stopping point and should be “invalid data”. For “invalid data”, it is automatically eliminated by setting a threshold (speed > 5 km/h). Through the above data filling and eliminating, a high-quality data basis can be provided for subsequent identification of habitats.

To test the effect of the spatiotemporal-density-based identification method on crested ibis’ roosting and foraging sites, the trajectory data of a crested ibis is used as the research object. The dataset includes 8760 data of crested ibis with numbered 22:CAFL003, 1245 data of crested ibis with numbered 20:CAFL005, and 312 data of crested ibis with numbered 20:CAFL001. Each data contains 8 bytes. The longest continuous period covered by the trajectory data is one year.

The hardware includes a processor Inteli7-11700, a main frequency 3.30 GHz, and memory 32 GByte. The software used includes Windows 10 operating system and QGIS 3.20 development environment, and Tianditu provided the base image.

### 2.3. Crested Ibis’s Stopping Points Detection Based on Spatiotemporal Density

The trajectory data of the crested ibis has spatial and temporal characteristics. According to both spatial and temporal density of the trajectory data, areas that the crested ibis visits with high frequency and stays for a long time can be identified. Therefore, combining the characteristics of spatial and temporal density, this paper proposes a spatial and temporal density-based habitat identification method.

This method is divided into two parts. The first step is to detect the stopping points of the crested ibis through thresholding, that is setting a spatial distance threshold; the second step is to accurately identify the habitats through the improved spatiotemporal clustering-based DBSCAN method.

#### 2.3.1. Stopping Points Detection by Spatial Thresholding

According to the loyalty to their nests and preference for foraging behavior [20], crested ibis’ stays in a stationary or small-scale moving state in both the overnight sites and foraging sites, that is, considered as a stopping state. Therefore, according to the spatial characteristics of longitude and latitude information in the trajectory data, this paper proposes a detection method for stopping points based on spatial thresholding.

Set the trajectory dataset as, and calculate the actual distance of stopping points (the trajectory data) in the map through latitude and longitude transformation, which is defined as:(1)$$T=\{p_{1},p_{2},\ldots ,p_{n}\}$$
(2)$$p_{i}=(p_{loi},p_{lai},p_{ti})$$
(3)$$A=\sin P_{lai}\sin P_{la(i-1)}$$
(4)$$B=\cos P_{lai}\cos P_{la(i-1)}\cos(P_{lo(i-1)}-P_{loi})$$
(5)$$D=111.12\cos(\frac{1}{A+B})$$

Among them, $p_{loi}$, $p_{lai}$ and $p_{ti}$ represent the longitude, latitude and time information of the crested ibis’s trajectory data point i, respectively; D represent the Euclidean distance of any two trajectory points.

According to the observation by the rangers of the Crested Ibis Reserve in Shanxi Province, the crested ibis is in a state of walking during the foraging period, and its activity range is about 50 m–200 m in the same feeding ground. Thus, by setting a spatial threshold, the trajectory data of crested ibis T can be classified into types, namely “high-frequency activity areas” and “low-frequency activity areas”. High-frequency activity areas $A_{d}$ and low-frequency activity areas $A_{s}$ are the sets of different trajectory data, which were defined as:(6)$$A_{d}={\{D}_{i}|\left|D_{i}\right|<50\}\;A_{s}={\{D}_{i}|\left|D_{i}\right|\geq 50\}$$

The yellow points on the map (Figure 1) represent all the information on the distribution of latitude and longitude coordinates of the crested ibis’s trajectories, located at 1 h intervals from 2 January 2015 to 2 January 2016. The scattered points (large distances between data points) represent low-frequency activity areas, indicating that the crested ibis visits these areas less frequently. These areas are considered moving points, occasional stopovers or noise spots for crested ibis and are not considered possible habitats for crested ibis. The clusters of data points with distances less than the spatial threshold represent the high-frequency activity area of crested ibis, indicating that they stop at similar locations. According to crested ibis loyalty to the nest and foraging behavior preference [21], the high-frequency activity areas mean crested ibis foraging or overnighting in these locations.

#### 2.3.2. Improved DBSCAN Method

Since crested ibises move less while sleeping, they stay in the same position for longer periods. When foraging, they will frequently move within a certain distance due to the influence of food distribution. Therefore, according to temporal characteristics, the stopping points (high-frequency activity area of crested ibis) can be divided into areas with high temporal density and low temporal density. High temporal density areas are those where the crested ibis stays for a long time continuously, which should be either their overnight sites or foraging sites. In contrast, the crested ibis stops discontinuously and irregularly at the low temporal density areas, which should only be the foraging sites. Therefore, this paper proposes an improved DBSCAN method based on spatiotemporal clustering to accurately identify their overnight and foraging sites.

The improved DBSCAN method is a density-based clustering method. The density of the target point is determined by counting the number of data contained within a certain radius of each target point’s neighborhood. It forms any shape of a data structure without knowing the number of targets in the detection range, which is suitable for identifying multiple stopping states of the crested ibis, such as roosting and foraging sites.

The set of the crested ibis’s stopping points T is divided into k subsets C with sufficient density according to two hyperparameters:(7)$$\begin{array}{l} T=\{C_{1},C_{2},\ldots ,C_{k}\}\\ \cap_{i=1}^{k}C_{i}=\emptyset \end{array}$$

For any data point $p_{i}\in T$, by setting the time period and spatial distance between the data point and sample points of its neighborhood, the clustering of stopping points is formed:(8)$$Dis\_s(p_{i},p_{j})\leq \varepsilon_{s}$$
(9)$$Dis\_t(p_{i},p_{j})\leq \varepsilon_{t}$$
(10)$$N(p_{i})=\{p_{i}\in T\}$$
of which $\varepsilon_{s}$ represents the spatial threshold, with a value of 50, and the unit is m; $\varepsilon_{t}$ represents the time threshold, with a value of 7, and the unit is h. The spatial and time threshold are empirical values that have been proven through experiments and field investigations.

If the density of the neighborhood of a sample point is greater than or equal to the threshold Td, the sample is regarded as a core point, and the sample points in the neighborhood, as well as the core point, belong to the same set. The sample points that cannot be connected within the range of any core point are noise points and do not belong to any set [11].

When a density reachability relationship is formed among sample points, it indicates that the samples belong to the same cluster. Since the DBSCAN algorithm marks the sample points by searching, the set of clusters connected with the maximum density derived by searching for the density is the habitat of crested ibis.

## 3. Results

### 3.1. High-Frequency Stopping Points Identification Results

Figure 2 shows the identification results of the crested ibis’s high-frequency stopping points based on the spatial threshold. Different colors represent different high-frequency stopping areas at different times from 2 January 2015 to 2 January 2016, marked from 1 to 6, respectively. These areas in the figure were visited by all the tagged crested ibis. Areas no 5–6 have hundreds of the crested ibis’s trajectory data, indicating these areas are the most commonly used place for crested ibis to stop, maybe their overnight and foraging sites. Areas no 1–4 are also areas with a higher spatial density of points but derived from less crested ibis’s trajectory data. These areas may also indicate sites occasionally used by crested ibises.

According to the remote sensing images in RGB format, the R-value and B-value of four stopping areas (1,4,5,6) are both less than 100, while the G-value ranges from 60 to 120, which indicates that the stopping area is green as a whole. This indicates that the stopping areas are green as a whole and are woodland with high vegetation coverage. Therefore, these areas are likely to belong to the places where all the tagged crested ibis sleep and forage. The RGB values of the no. 2 and no. 3 stopping areas are not in the above range, indicating that the vegetation coverage in these two areas is low. These areas do not correspond to the ibis’ regular overnight stays and may only be feeding grounds of all the tagged crested ibis. However, the location of crested ibises sleeping and foraging in the woodland cannot be accurately determined only by the stopping point detection method.

### 3.2. Identification Results of Crested Ibis’s Overnight Sites

Figure 3 shows the result of identifying stopping areas using the improved spatiotemporal clustering based on the DBSCAN method. Red points represent detected stopping points with high spatial and temporal density. The white points are the centroids of many consecutive stopover points, which are used to characterize the entire stopover area of the tagged crested ibis.

In Figure 3, the areas with more red points indicate the common overnight places for the crested ibis. Areas with fewer red stops indicate that crested ibises occasionally discontinuously stop, possibly as their temporary overnight sites. The stopping points are distributed in an area, which is consistent with the living habit of the crested ibis, which is changing positions nearby to protect themselves when sleeping. According to the remote sensing images in RGB format, these areas are all concentrated in woodland with high vegetation coverage. These findings coincide with the identified types of the crested ibis’s overnight habitats.

### 3.3. Identification Results of Foraging Sites

The identification results of crested ibis’s habitats based on spatial and temporal density have shown in Figure 4. The overlapping areas of red stopping points and other color stopping points are characterized by high spatiotemporal density, such as red and blue overlapping areas, and red and purple overlapping areas, indicating that all the tagged crested ibis have been staying here for a long time. These areas indicate where crested ibises spend their nights.

The other stopping areas without red stopping points are mainly distributed in farmland and rivers. The temporal distance between stopping points in such areas is relatively large and characterized as “high spatial density” and “low temporal density”, indicating that the tagged crested ibis visits such areas regularly or irregularly at certain time intervals. This area is a feeding ground for crested ibis. Figure 4 shows that all the tagged crested ibis’ forage for food in or around the overnight sites, which is also consistent with its living habits. The area with only red stopping points has only a few stop points with high spatial and temporal density, which means that crested ibis’ stay there for a long time, but with low frequency. This may be an occasional overnight place for the tagged crested ibis, or it may be a place that meets the conditions for crested ibis’ future habitat migration.

### 3.4. Verify Habitat Compatibility of Identified Overnight and Foraging Sites

To verify the effectiveness of the proposed method, a field investigation was conducted by six people for eight days in 2021. The results of identified overnight and foraging sites are shown in Table 2.

In Table 2, the large trees such as *Pinus Massoniana*, *Platycladus orientalis*, and *Quercus acutissima* are widely distributed in the overlapped areas (Area no. 1 and no. 4–6). These areas are all woodland, where crested ibis sleep and feed. The non-overlapping areas (no. 2 and no. 3) are rivers and farmland, both of which are places for crested ibis’ to feed. The experimental results show that the crested ibis’s overnight sites and foraging sites found in the field investigation can be correctly identified by the proposed method in this paper.

## 4. Discussion

In the early stage of crested ibis protection, the number of wild crested ibis nests was small, and the crested ibis had few and very stable overnight habitats, and the distribution of their habitats could be accurately grasped through daily patrols [22]. In the past ten years, with the increase in crested ibis’, their distribution area has rapidly expanded to more than ten places outside Yangxian County. It has been difficult to accurately monitor the habitat of the crested ibis by conventional research methods [3], which brings great challenges to the protection and management of the crested ibis’s habitats. The development of wearable GPS transmitters provides an effective method for real-time monitoring of crested ibis.

In Figure 3, the seven crested ibis overnight sites are scattered on the map at a certain distance. This is because the crested ibis has strong territorial characteristics during the breeding season and will drive other birds away by singing and flying. Among them, the distance between two ibis overnight sites is much smaller than that of other sites. This may be because the crested ibis has a tendency to nest in clusters, and the crested ibis’s habitats are also spreading to the periphery with the increase in the crested ibis population, the change of habitat, and the limitation of environmental capacity [23].

The crested ibis’s feeding grounds are mainly concentrated in the woodlands around the overnight places, farmland and rivers (Figure 4). The first situation may be that the crested ibis, during the breeding season, is carrying out activities such as hatching and brooding. At this time, the foraging grounds of crested ibis’ are relatively stable, and they are usually distributed in the area near the center of the night. During the wintering period and late breeding period, the nearby food resources are relatively scarce, and the crested ibis foraging grounds are scattered, usually far away from the overnight places [24].

In addition, based on the crested ibis’s trajectory data, the habitat selection preferences of crested ibis can be determined, providing decision support for crested ibis’s habitat protection and selection. At the same time, it can also provide data support for future research on the living habits and behaviors of crested ibis’ in different periods.

## 5. Conclusions

An automatic identification method of the crested ibis’s habitats based on spatiotemporal density detection is proposed in this paper. According to the spatial and temporal characteristics of the trajectory data, the method can quickly and accurately identify their overnight and foraging sites. Through the combination of qualitative analysis and on-site investigation, the effectiveness of this method for automatically identifying crested ibis’ habitats is analyzed. Through remote sensing images and field investigations, it was concluded that woodlands with large arbor trees are common overnight places for the crested ibis. In contrast, other woodlands, farmland, and river areas are common foraging sites.

Meanwhile, the method can also find overnight places and foraging places that are not easily observed by traditional methodologies (field trips and surveys). Therefore, the method in this paper can quickly and accurately identify common habitats and potential habitats of crested ibis’, and then provide decision support for the crested ibis’s habitat protection and habitat search.

## Figures and Tables

**Figure 1 animals-12-02220-f001:**
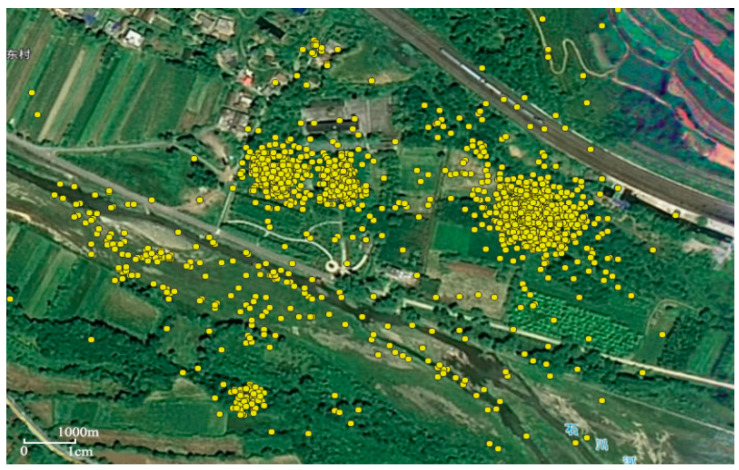
Distribution of latitude/longitude coordinates of crested ibis’s trajectory data. Each yellow point represents the crested ibis’s trajectory data at 1 h intervals from 2 January 2015 to 2 January 2016.

**Figure 2 animals-12-02220-f002:**
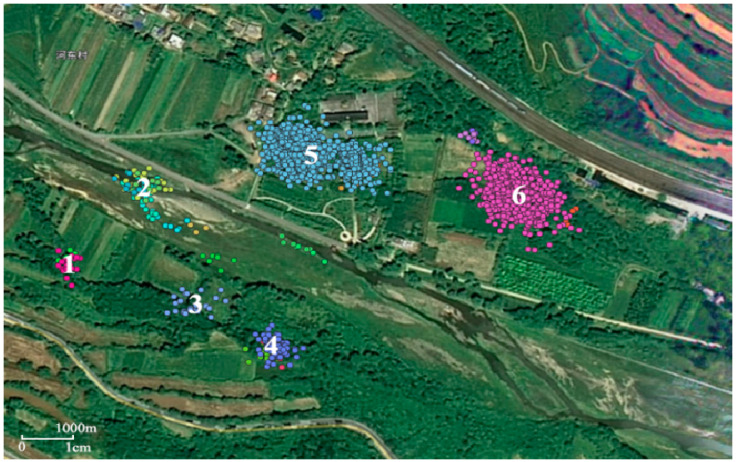
Identification results of the crested ibis’s high-frequency stopping points based on the spatial threshold. Different colors represent different high-frequency stopping areas at different times from 2 January 2015 to 2 January 2016, marked from 1 to 6, respectively.

**Figure 3 animals-12-02220-f003:**
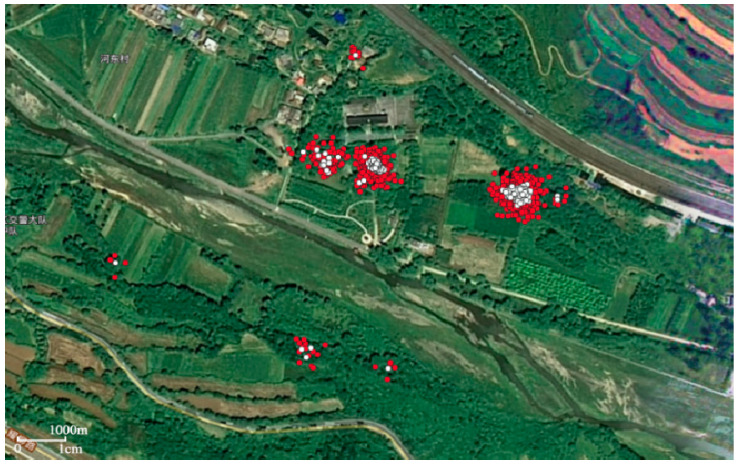
Result of identifying stopping areas using the improved DBSCAN method. Red points represent detected stopping points with high spatial and temporal density. White points are the centroids of many consecutive stopover points.

**Figure 4 animals-12-02220-f004:**
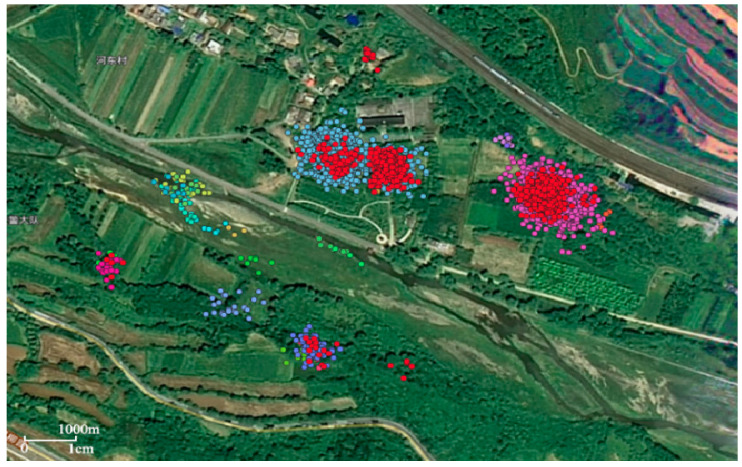
Identification results of crested ibis’s habitats based on spatial and temporal density. Track data points of all the tagged crested ibis in colors other than red represent different types of stops. The overlapping areas of red stopping points and other color stopping points indicate crested ibis’ overnight sites. Only the red stop point area indicates the occasional nightspot of crested ibis. The stopping areas without red stopping points indicate crested ibis’ foraging sites.

**Table 1 animals-12-02220-t001:** Example of crested ibis’s trajectory data.

Terminal	22:CAFL003	22:CAFL003	22:CAFL003
Time	2020/3/31 6:00	2020/3/31 8:00	2020/3/31 10:00
Longitude	E108.82488	E108.82478	E108.82490
Latitude	N35.04734	N35.04731	N35.04737
Speed	0	0	0
Altitude	183.3	849.1	178.5
Heading direction	9.5	163.9	0
Accuracy	A	A	B
Time	2020/3/31 6:00	2020/3/31 8:00	2020/3/31 10:00

**Table 2 animals-12-02220-t002:** Crested ibis overnight and foraging sites identified by field investigation.

Numbering	Type	Overnight Sites	Foraging Sites
1	woodland	yes	yes
2	river	no	yes
3	Farmland	no	yes
4	woodland	yes	yes
5	woodland	yes	yes
6	woodland	yes	yes

## Data Availability

These data are available upon request from the corresponding author.

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
