# Peer review of "An Automatic Identification Method of Crested Ibis (Nipponia nippon) Habitat Based on Spatiotemporal Density Detection"

_animals, 2022, doi:10.3390/ani12172220_

Round 1
Reviewer 1 Report
Dear Authors,
the paper must be improved to published in Animals. My remarks line by line:
Line 2: automatic, not Automatic
Line 16: add (Temminck, 1835) to Latin name of the species , the same do in line 39.
Line 32: it was, not is is. The article is finished.
Line 42 - put a space between species and [8,9].
Line 49 - please change it for the whole text: two large space between ibis' s
Line 49 - delete &
Line 55 - delete and, because short name etc. include "and"
Line 56 - two large space after on-site
Line 89 and 90 - put a space between numbers and m
Line 93 - what do you mean writing "construction land"? It is not understandable.
Line 104 - put a space after table and before this line
Line 148 - each figure put in the middle of width of pages
Line 148 - each figure in the text should have a legend, or at least more detailed title. Without the text, we don;'t know nothing what is is in the figure. Each figure should have a clear scale! with m or km.
Line 188 - put 'TU' inside the previous line - make these lines thicken
Line 196 - not from the figure, but in
Line 204 and 226 - the same first part of paragrahs. It should not look like that.
Line 208 - write the name of the species in Latin in Italics
Line 219 - add the legend or more detailed title. Do it consequently the same for the whole text
Line 220-221 - try to substitue overnight in one place or built this sentence in a different way
Line 236 - I really can't see the differenced between figure 3 and 4
Line 244 - These are only Conclusions. You need to add another separate chapter "Disscussion". It is the most important chapter in each article.
Good luck
Author Response
Responsed to Reviewer #1 Comments:
First of all, thank you so much for your efforts on offering valuable comments and helpful suggestions. We have carefully addressed your comments and revised our manuscript accordingly. Our responses (in normal font with blue) to your constructive criticisms (in italics) are as follows:
Comment 1: Line 2: automatic, not Automatic
Line 16: add (Temminck, 1835) to Latin name of the species , the same do in line 39.
Line 32: it was, not is is. The article is finished.
Line 42 - put a space between species and [8,9].
Line 49 - please change it for the whole text: two large space between ibis' s
Line 49 - delete &
Line 55 - delete and, because short name etc. include "and"
Line 56 - two large space after on-site
Line 89 and 90 - put a space between numbers and m
Line 93 - what do you mean writing "construction land"? It is not understandable.
Line 104 - put a space after table and before this line
Line 148 - each figure put in the middle of width of pages
Response 1: We appreciate your comments and suggestions on our manuscript very much. According to your suggestion, we have revised the manuscript and marked the revised content in blue.
- We have used "automatic" instead of "Automatic". (Page1, Line 2)
- We have added "Temminck, 1835" to Latin name of the species. The Latin name changed to"Nipponia nippon Temminck, 1835". (Page 1, Line 22 and Line 39)
- We have used "was" instead of "is". (Page1, Line 2)
- We have put a space between species and [8,9]. (Page1, Line 43)
- We have used "ibis's" instead of " ibis’s" in the whole manuscript.
- We have deleted"&". (Page 2, Line 50)
- We have deleted"and". (Page 2, Line 56)
- We have used "field surveying" instead of "in-person on-site surveying". (Page 2, Line 56)
- We have put a space between numbers and m. "500 m", "2900 m", "50 m", "200 m", "79 mm", "23 mm", "30 mm", "32 g". (Page 2, Line 89; Page 3, Line 97; Page 4, Line 144)
- We have used "rice paddies, reservoirs, rivers, and farmlands in valleys" instead of "forestland, farmland, water body, and construction land". (Page 2, Line 93)
- We have put a space before and after Table (Page 3, Line 106)
- Each figure hasbeen put in the middle of width of pages. (Page 4, Line 159; Page 6, Line 216; Page 7, Line 237; Page 7, Line 255)
Comment 2: Line 148 - each figure in the text should have a legend, or at least more detailed title. Without the text, we don;'t know nothing what is is in the figure. Each figure should have a clear scale! with m or km.Line 219 - add the legend or more detailed title. Do it consequently the same for the whole text
Response 2: Thanks for your constructive suggestions which are critical for us to better illustrate the figures. We have revised the captions of all figures in the manuscript. I believe the revised captions could be self-explanatory. In Figures, the scale of the map is 1:10000, that is, 1cm in the map means the actual distance is 1000m. (Page 4, Line 160; Page 6, Line 217; Page 7, Line 237; Page 7, Line 255)
Comment 3: Line 188 - put 'TU' inside the previous line - make these lines thicken
Line 196 - not from the figure, but in
Line 204 and 226 - the same first part of paragrahs. It should not look like that.
Line 208 - write the name of the species in Latin in Italics
Line 236 - I really can't see the differenced between figure 3 and 4
Line 220-221 - try to substitue overnight in one place or built this sentence in a different way
Response 3: Thank you very much for your suggestion.
- We modify this sentence to " and Tianditu provided the base image." (Page 6, Line 205)
- We have used "in" instead of "from". (Page6, Line 209)
- We have revised the paragraphs. (Page6-7, Line 219-242)
- We have revised the name of the species in Latin in Italics in the whole manuscrip
- I'm sorry it was my mistake. We have revised figures3 and 4.
- We have changedthis sentence to "In addition, due to the influence of the external environment, crested ibis will change the place of the night when they sleep overnight. Therefore, multiple crested ibis’s overnight sites will appear more continuously within a certain range." (Page 7, Line 240-242)
Comment 4: Line 244 - These are only Conclusions. You need to add another separate chapter "Discussion". It is the most important chapter in each article.
Response 4: Thank you very much for your suggestion which is critical for us to better illustrate the contribution of our paper. According to your suggestion, we have added the Discussion Section in the manuscript. (Page 9-10, Line 284-313)
Expressions of the whole manuscript are checked and revised to our best. Language modifications have also been made by native English speakers.Hope the manuscript could be more readable and understandable.

Reviewer 2 Report
This article descripted a process of identifying key micro-habitats (namely the
roosting and foraging sites) of the crested ibis. As a methodology experiment, this
article fails to show its justification, as well as lack of a quantitative comparison with
traditional methodology (i.e., field visits and surveys) to show the value of this new
method. Major defects including the definitions and the usage of “spatial density”
and “temporal density”, which were the key elements of this study:
1. In line 144-147 stated “the scattered points represent low-frequency activity
areas, indicating that crested ibis visits these areas less frequently, which are
considered non-habitation sites or noise points. The red circled areas represent
high-frequency activity areas, indicating stopping sites, meaning foraging for
food or sleeping overnight during this period.” In line 151-154 stated “High
temporal density areas are those where crested ibis stays for a long time
continuously, which should be either their habitats or foraging site; while
crested ibis stops discontinuously and irregularly at the low temporal density
areas, which should only be the foraging sites.” Later in Line 197-201 stated
“high spatial density area are common habitats, while low spatial density area
are crested ibis occasionally, discontinuously, and irregularly stops, which may
be their secondary habitats and foraging sites.” This article used three terms to
descript the same distribution patterns of GPS positions and gave different
interpretations (non-habitat vs secondary habitat vs less function habitat) to
different distribution patterns (dense vs scattered), which is not precise enough.
Also, no confirmation of these interpretations were provided in the article.
2. In line 138-139 stated “Since the activity range in the overnight roosting areas
and foraging sites is 50m-200m.” This statement requires reference because it
later been used as the spatial threshold in the algorithm for GPS positions
screening (see line 168-169).
3. For the time threshold of 7 hours (line 168-169) also need to provide reference.
4. In line 207-209 stated “The field investigation conducted in 2021 also found that
large trees such as Pinus Massoniana are widely distributed in this area, a
common habitat and foraging site for crested ibis. This conclusion is consistent
with the results obtained via this method.” If people already know this kind of
trees are common habitat and forage site for this bird, why need to use a
complicate way to prove it again? Also, I can’t find the data from this study.
5. A minor point: there is no yellow circles in Fig. 4 (see line 241)
Author Response
Responsed to Reviewer #2 Comments:
We would like to thank you for careful and thorough reading of this manuscript and for the thoughtful comments and constructive suggestions, which help to improve the quality of this manuscript. We have modified the manuscript accordingly, and the detailed corrections are listed below point by point (the reviewer’s comments are in italics and the authors’ responses are in normal font with blue):
Comment 1: This article descripted a process of identifying key micro-habitats (namely the roosting and foraging sites) of the crested ibis. As a methodology experiment, this article fails to show its justification, as well as lack of a quantitative comparison with traditional methodology (i.e., field visits and surveys) to show the value of this new method. Major defects including the definitions and the usage of “spatial density” and “temporal density”, which were the key elements of this study:
Response 1: Thank you very much for your suggestion which is critical for us to better illustrate the contribution of our paper. According to your suggestion, we have revised the manuscript.
The exploration of the characteristics and selection rules of habitats such as foraging sites and overnight sites can provide a scientific basis for evaluating the living conditions of crested ibis so as to formulate appropriate protection and management measures. The traditional methods (field visit and survey) are used to monitor crested ibis living areas through daily inspections of crested ibis key activity areas, breeding monitoring, and field surveys of nocturnal habitats.
However, with the rapid growth of its population and sharp expansion of commonly used habitats, monitoring crested ibis’s stopping points (such as overnight sites and foraging sites) by field surveying faces significant challenges, including heavy workload, time-consuming and labor-intensive. In addition, it is impossible to obtain the trajectory of their real-time movements and potential stopping points, resulting in failure to protect their habitats in a timely and effective manner. In addition, after nearly 20 years of research on crested ibis in Shanxi Province, the team members found that in addition to the commonly used habitats, the crested ibis gradually has some occasionally used habitats, which are also difficult to find through traditional methods. The development of wearable GPS transmitters in navigation and positioning systems provides an efficient approach for the real-time acquisition of trajectory data. Based on the trajectory data of crested ibis, analyzing the resource status of the habitat has great significance for the protection of their habitats.
Comment 2: In line 144-147 stated “the scattered points represent low-frequency activity areas, indicating that crested ibis visits these areas less frequently, which are considered non-habitation sites or noise points. The red circled areas represent high-frequency activity areas, indicating stopping sites, meaning foraging for food or sleeping overnight during this period.” In line 151-154 stated “High temporal density areas are those where crested ibis stays for a long time continuously, which should be either their habitats or foraging site; while crested ibis stops discontinuously and irregularly at the low temporal density areas, which should only be the foraging sites.” Later in Line 197-201 stated “high spatial density area are common habitats, while low spatial density area are crested ibis occasionally, discontinuously, and irregularly stops, which may be their secondary habitats and foraging sites.” This article used three terms to descript the same distribution patterns of GPS positions and gave different interpretations (non-habitat vs secondary habitat vs less function habitat) to different distribution patterns (dense vs scattered), which is not precise enough. Also, no confirmation of these interpretations were provided in the article.
Response 2: Thank you very much for your careful review and we are extremely sorry for those non-uniform expressions in the manuscript. According to your suggestion, we have revised the description of this part in the manuscript. We unify the interpretation of dense versus scattered distribution patterns. We also use the concept of habitat uniformly and give an explanation.
- Line 144-147 were modified to “From the distribution of latitude/longitude coordinates shown in Figure 1, the yellow points represent the crested ibis’s trajectory data at the current moment. The scattered points (large distances between data points) represent low-frequency activity areas, indicating that crested ibis visits these areas less frequently. These areas are considered moving points, occasional stopovers or noise spots for crested ibis and are not considered possible habitats for crested ibis. The clusters of data points with distances less than the spatial threshold represent the high-frequency activity area of crested ibis, indicating that they stop at similar locations. According to crested ibis loyalty to the nest and foraging behavior preference, the high-frequency activity areas mean crested ibis foraging or overnighting in these locations.” (Page 3, Line 149-158)
- Line 151-154were modified to “Since crested ibises move less while sleeping, they stay in the same position for longer periods. When foraging, they will frequently move within a certain distance due to the influence of food distribution. Therefore, according to temporal characteristics, the stopping points (high-frequency activity area of crested ibis) can be divided into areas with high temporal density and low temporal density. High temporal density areas are those where crested ibis stays for a long time continuously, which should be either their overnight sites or foraging sites. In contrast, crested ibis stops discontinuously and irregularly at the low temporal density areas, which should only be the foraging sites.” (Page 4, Line 163-170)
- Line 197-201were modified to “Figure 2 shows the identification result of crested ibis’s high-frequency stopping points based on the spatial threshold. Different colors in the figure represent different high-frequency activity areas, marked from 1 to 6, respectively. It can be seen in Figure 2 that there are dozens of crested ibis’s trajectory data in the No. 5 and No. 6 areas. This suggests that these areas are the most commonly used place for crested ibis to stop, maybe their overnight and foraging sites. The difference is that the No. 1, No. 2, No. 3, and No. 4 areas are also distribution areas with higher spatial density, but there are fewer crested ibis’s trajectory data. These indicate that crested ibises occasionally stop in a place that may also be their overnight sites and foraging sites.” (Page 6, Line 207-215)
Comment 3: In line 138-139 stated “Since the activity range in the overnight roosting areas and foraging sites is 50m-200m.” This statement requires reference because it later been used as the spatial threshold in the algorithm for GPS positions screening (see line 168-169). For the time threshold of 7 hours (line 168-169) also need to provide reference.
Response 3: Thank you very much for your suggestion. According to the observation by the rangers of the Crested Ibis Reserve in Shanxi Province, the crested ibis is in a state of walking during the foraging period, and its activity range is about 50 m- 200 m in the same feeding ground. Experiments show that when the time threshold is 7h, the judgment of the crested ibis habitat is correct. Therefore, the spatial and temporal thresholds are empirical values proven through experiments and field investigations. (Page 4, Line 142-144; Page 5, Line 185-187)
Comment 4: In line 207-209 stated “The field investigation conducted in 2021 also found that large trees such as Pinus Massoniana are widely distributed in this area, a common habitat and foraging site for crested ibis. This conclusion is consistent with the results obtained via this method.” If people already know this kind of trees are common habitat and forage site for this bird, why need to use a complicate way to prove it again? Also, I can’t find the data from this study.
Response 4: We are extremely sorry for those non-uniform expressions in the manuscript. We have revised the descript of the results of the field investigation. Based on the analyzed crested ibis habitats, we conducted a field investigation and found that the crested ibis habitats are all woodlands, and large trees such as Pinus Massoniana, Platycladus orientalis, and Quercus acutissima are widely distributed. The correctness of the proposed method in this manuscript is verified by field investigation. Therefore, this manuscript can provide a convenient technical means for the real-time tracking of crested ibis trajectory and the selection and protection of crested ibis’s habitats.
“To verify the effectiveness of the proposed method, the field investigation was conducted by six people for eight days in 2021. The results in Table 2 show that the large trees such as Pinus Massoniana, Platycladus orientalis, and Quercus acutissima are widely distributed in the overlapped areas (No. 1, No. 4, No. 5, and No. 6). These areas are all woodland, where crested ibis sleep and feed. The non-overlapping areas (No. 2 and No. 3) are rivers and farmland, both of which are the places for crested ibis to feed. Field inspection found that the results of crested ibis’s overnight sites and foraging sites obtained by the method in this paper were correct.”. (Page 8, Line 272-279)
Comment 5: A minor point: there is no yellow circles in Fig. 4 (see line 241)
Response 5: Thank you very much for your suggestion. We have deleted the “yellow circles in Fig. 4” and revised the sentence. (Page 8, Line 272-283)
Expressions of the whole manuscript are checked and revised to our best. Language modifications have also been made by native English speakers.Hope the manuscript could be more readable and understandable.

Reviewer 3 Report
The authors have presented an interesting research project wherein they have shown how the use of backpack transmitters can help in understand ng the spatial ecology of an endangered species, the crested ibis. The conservation implications of the study are important.
However, the language requires a careful edit. In many places, it hampers the flow of the paper. The discussion is extremely short and inappropriate. Recommend the authors compare with other studies of waterfowl to give a more international perspective to the paper. Also give specifics of the backpack transmitter so we can know the company, the system used, etc.
Author Response
Response to Reviewer #3 Comments:
We would like to thank you for careful and thorough reading of this manuscript and for the thoughtful comments and constructive suggestions, which help to improve the quality of this manuscript. We have modified the manuscript accordingly, and the detailed corrections are listed below point by point (the reviewer’s comments are in italics and the authors’ responses are in normal font with blue):
Comment: The authors have presented an interesting research project wherein they have shown how the use of backpack transmitters can help in understand the spatial ecology of an endangered species, the crested ibis. The conservation implications of the study are important. However, the language requires a careful edit. In many places, it hampers the flow of the paper. The discussion is extremely short and inappropriate. Recommend the authors compare with other studies of waterfowl to give a more international perspective to the paper. Also give specifics of the backpack transmitter so we can know the company, the system used, etc.
Response: Thank you very much for your suggestion which is critical for us to better illustrate contribution of our paper. According to your suggestion, we have added the Discussion Section, revised the Conclusion Section, and added the specifics of the backpack transmitter.
- “The crested ibis’s trajectory was tracked by using the HQBG3621L backpack-style tracker (provided by Hunan Global Letter Technology Co., Ltd., China), which is about 79 mm long, 23 mm wide, 30 mm high, and weighs about 26~32 g. The tracker collects crested ibis’s trajectory data in real- time through the combination of Beidou and GPS. The tracker has a battery life of up to five years when the average sunshine duration is greater than 1 hour. ”(Page 2-3, Line 95-100)
- The Discussion Section has been added, “In the early stage of crested ibis protection, the number of wild crested ibis nests was small, and the crested ibis had few and very stable overnight habitats, and the distribution of their habitats could be accurately grasped through daily patrols. In the past ten years, with the increase in crested ibis, their distribution area has rapidly expanded to more than ten places outside Yangxian County. It has been difficult to accurately monitor the habitat of crested ibis by conventional research methods, which brings great challenges to the protection and management of crested ibis’s habitats. The development of wearable GPS transmitters provides an effective method for real-time monitoring of crested ibis.
- In Figure 3, the seven crested ibis’s overnight sites are scattered on the map at a certain distance. This is because crested ibis has strong territorial characteristics during the breeding season and will drive other birds away by singing and flying. Among them, the distance between two ibis overnight sites is much smaller than that of other sites. This may be because the crested ibis has a tendency to nest in clusters, and the crested ibis’s habitats are also spreading to the periphery with the increase of the crested ibis population, the change of habitat, and the limitation of environmental capacity.
- Crested ibis’s feeding grounds are mainly concentrated in the woodlands around the overnight places, farmland and rivers (Figure 4). The first situation may be that the crested ibis, during the breeding season, is carrying out activities such as hatching and brooding. At this time, the foraging grounds of crested ibis are relatively stable, and they are usually distributed in the area near the center of the night. During the wintering period and late breeding period, the nearby food resources are relatively scarce, and the crested ibis foraging grounds are scattered, usually far away from the overnight places .
- In addition, based on the crested ibis’s trajectory data, the habitat selection preferences of each crested ibis species can be determined, including wild crested ibis and imported crested ibis, providing decision support for crested ibis’s habitat protection and selection. At the same time, it can also provide data support for future research on the living habits and behaviors of each crested ibis in different periods.” (Page 8-9, Line 284-313)
- The Conclusion Section has been revised, “An automatic identification method of crested ibis’s habitats based on spatiotemporal density detection is proposed in this paper. According to the spatial and temporal characteristics of the trajectory data, the method can accurately identify their overnight and foraging sites in real-time. Through the combination of qualitative analysis and on-site investigation, the effectiveness of this method for automatically identifying crested ibis’ habitats is analyzed. Through remote sensing images and field investigations, it was concluded that woodlands with large arbor trees are common overnight places for crested ibis. In contrast, other woodlands, farmland, and river areas are common foraging sites.
- Meanwhile, the method can also find overnight places and foraging places that are not easily observed by traditional methodologies (field trips and surveys). Therefore, the method in this paper can identify common habitats and potential habitats of crested ibis in real-time, and then provide decision support for crested ibis’s habitat protection and habitat search. ” (Page 8-9, Line 284-313)
- Expressions of the whole manuscript are checked and revised to our best. Language modifications have also been made by native English speakers.Hope the manuscript could be more readable and understandable.

Round 2
Reviewer 1 Report
Dear Authors, I have some small remarks to the text, as well as graphic issues that have to be changed.
Line 40 - 'and also and' - is it correct?
Quality of each figure need to be changed. Please generate figures in better resolution! As well as add a scale bar to each figure. It is not professional to show an ortophotomap without context of reality!
Line 159 - figure one - work on better quality, add scale bar
Line 216 - Figure 2 - work on better quality, add scale bar
Line 224 - no. instead of No.
Line 237 - Figure 3 - work on better quality, add scale bar
Line 254 - Figure 4 - work on better quality, add scale bar
Line 270 - table 2 - why this table is in different font?? Change it into the same font like whole text.
Line 275, 276 and in other places in the text - no. instead of No. It is not a name.
Line 338 - why double -?. Add a dot at the end of the sentence.
Good luck |
Author Response
Responsed to Reviewer #1 Comments:
We would like to thank you for careful and thorough reading of this manuscript and for the thoughtful comments and constructive suggestions, which help to improve the quality of this manuscript. We have modified the manuscript accordingly, and the detailed corrections are listed below point by point (the reviewer’s comments are in italics and the authors’ responses are in normal font with blue):
Comment 1: Dear Authors, I have some small remarks to the text, as well as graphic issues that have to be changed.
Line 40 - 'and also and' - is it correct?
Line 270 - table 2 - why this table is in different font?? Change it into the same font like whole text.
Line 275, 276 and in other places in the text - no. instead of No. It is not a name.
Line 338 - why double -?. Add a dot at the end of the sentence.
Response 1: We appreciate your comments and suggestions on our manuscript very much. According to your suggestion, we have revised the manuscript and marked the revised content in blue.
- We have used "and" instead of "and also and". (Page1, Line 40)
- We have change Table 2 into the same font with Table.(Page 8, Line 270)
- We have used "" instead of "No."in the whole manuscript.
- We haverevised the section of Funding and added a dot at the end of the sentence. “The research was supported by Fundamental Research Funds for the Central Public Welfare Research Institutes: Visual Analysis of Animal Trajectory Monitoring Data-Taking Crested Ibis (Nipponia nippon) as an Example (CAFYBB2021SY008).” (Page 9, Line 333-335)
Comment 2: Quality of each figure need to be changed. Please generate figures in better resolution! As well as add a scale bar to each figure. It is not professional to show an ortophotomap without context of reality!
Line 159 - figure one - work on better quality, add scale bar
Line 216 - Figure 2 - work on better quality, add scale bar
Line 224 - no. instead of No.
Line 237 - Figure 3 - work on better quality, add scale bar
Line 254 - Figure 4 - work on better quality, add scale bar
Response 2: Thanks for your constructive suggestions which are critical for us to better illustrate the figures.The quality of each figure has been changed with better resolution. Meanwile, we have added the scale bar to each figure.
Reviewer 2 Report
see attachment.

Author Response
Responsed to Reviewer #2 Comments:
We would like to thank you for hard work. We have modified the manuscript accordingly, and the detailed corrections are listed below point by point (the reviewer’s comments are in italics and the authors’ responses are in normal font with blue):
Comment 1: Comments to the revised manuscript in title of “An automatic identification method of crested ibis (Nipponia nippon) habitat based on spatiotemporal density detection.” After reading the revised version, I still cannot recommend the journal to publish this manuscript at its present status. My major comments are:
- The application of this modified DBSCAN algorithm in wildlife protection is questionable. As mentioned in Line 316-318 “According to the spatial and temporal characteristics of the trajectory data, the method can accurately identify their overnight and foraging sites in real-time.” And in Line 325-328 “the method in this paper can identify common habitats and potential habitats of crested ibis in real-time, and then provide decision support for crested ibis’s habitat protection and habitat search.” This study aimed to develop a quick (real-time) assessment method for managers to be able to fast provide necessary protection to the crested ibis. But, to perform such an analysis requires enough data (GPS locations in this case), and I cannot image how this kind of analysis can be done “in real time.”
Response 1: Thank you very much for your suggestion and we are sorry for the about the inappropriate description of “in real-time”. According to your suggestion, we have revised the manuscript. Compared with field survey methods, this paper expects to be able to identify crested ibis habitats more quickly and easily. Therefore, we modified "in real-time" to "quickly".
“According to the spatial and temporal characteristics of the trajectory data, the method can quickly and accurately identify their overnight and foraging sites.” (Page 9, Line 313-315)
“Therefore, the method in this paper can quickly and accurately identify common habitats and potential habitats of crested ibis, and then provide decision support for crested ibis’s habitat protection and habitat search.” (Page 9, Line 322-324)
Comment 2: On the other hand, this modified DBSCAN method might be useful in understanding animal’s microhabitat selection, however this study didn’t provide necessary information to show its accuracy. For example, how accurate is to add temporal and spatial thresholds as screening criteria for resting sites or feeding sites from the huge GPS location’s data set? Whether separation of the whole data set into nighttime and daytime location sets will increase the accuracy?
Response 2: Thank you very much for your careful review. In order to verify the effectiveness of our method for crested ibis habitat identification, we compared the identification results of our method with the results of the most recognized field survey methods. The experimental results show that the crested ibis habitat found in the field investigation can be correctly identified by the method in this paper. Therefore, it is considered feasible to use the temporal and spatial thresholds as screening criteria for resting sites or feeding sites. It is also an interesting idea to divide the entire dataset into nighttime and daytime location sets for habitat identification, but the split time of the dataset may not be easy to determine. In this paper, it would be simpler to perform habitat identification directly from the entire dataset.
Comment 3: Without providing necessary information, such as the number of birds used in this study, their genders, their tracking period and whether it covered breeding season or different seasons…etc., and perform proper analysis and show in Result section, it is not possible to judge the rationality of the whole discussion from Line 294 to Line 308.
Response 3: Thank you very much for your suggestion. Since adult crested ibis are difficult to capture, trackers were used in crested ibis juveniles in this study. The tracker for each crested ibis from birth until the crested ibis died or the tracker was unable to obtain data, and the trajectory data covered the breeding season. Due to the small number of adult crested ibis, the annual number of nesting and larvae is also low. In addition, bird experts believe that long-term wear of the transmitter has an effect on young birds. Therefore, in order to protect the growth of crested ibis chicks, three chicks were selected for tracking in this experiment. Since crested ibis have cluster characteristics, the trajectory data of birds can well represent the movement trajectory of the population. Moreover, during the long-term operation, due to the lack of power supply of the tracker, extreme weather, obstructions and other abnormal conditions, the collected data has problems such as data loss and low accuracy. Therefore, this paper takes the complete trajectory data of a crested ibis as an example to verify the applicability and correctness of the method, in order to provide technical support for the identification and protection of crested ibis habitats.
Reviewer 3 Report
Dear Editor,
I have now read the revised paper on the automatic identification of habitat preferences in crested ibis using spatiotemporal density detection by telemetry. The study is applicable to any and all species that are equipped with transmitters and hence can appeal to a larger audience for its usefulness and practicality. Hence, recommend accepting and the few glitches that exist will I suspect get ironed out in the editorial process.
Author Response
Response to Reviewer #3 Comments:
Thank you so much for your efforts on offering valuable comments and helpful suggestions. We have carefully addressed your comments and revised our manuscript accordingly. Our responses (in normal font with blue) to your constructive criticisms (in italics) are as follows:
Comment: I have now read the revised paper on the automatic identification of habitat preferences in crested ibis using spatiotemporal density detection by telemetry. The study is applicable to any and all species that are equipped with transmitters and hence can appeal to a larger audience for its usefulness and practicality. Hence, recommend accepting and the few glitches that exist will I suspect get ironed out in the editorial process.
Response: Thank you very much for your suggestion which is critical for us to better illustrate contribution of our paper. When given the opportunity, we will carefully revised the manuscript during the editorial process.